# Effects of Luteolin on Biofilm of *Trueperella pyogenes* and Its Therapeutic Effect on Rat Endometritis

**DOI:** 10.3390/ijms232214451

**Published:** 2022-11-21

**Authors:** Luyao Zhang, Yitong Cai, Lishuang Li, Chen Chen, Hanyu Zhao, Zehui Zhang, Yaochuan Liu, Yingyu Wang, Chunlian Tian, Mingchun Liu

**Affiliations:** Key Laboratory of Livestock Infectious Diseases in Northeast China, Ministry of Education, College of Animal Science and Veterinary Medicine, Shenyang Agricultural University, Dongling Road 120, Shenyang 110866, China

**Keywords:** luteolin, *Trueperella pyogenes*, biofilm, endometritis, flavonoids

## Abstract

*Trueperella pyogenes* is an opportunistic pathogen that causes suppurative infections in animals. The development of new anti-biofilm drugs will improve the current treatment status for controlling *T. pyogenes* infections in the animal husbandry industry. Luteolin is a naturally derived flavonoid compound with antibacterial properties. In this study, the effects and the mechanism of luteolin on *T. pyogenes* biofilm were analyzed and explored. The MBIC and MBEC of luteolin on *T. pyogenes* were 156 μg/mL and 312 μg/mL, respectively. The anti-biofilm effects of luteolin were also observed by a confocal laser microscope and scanning electron microscope. The results indicated that 312 μg/mL of luteolin could disperse large pieces of biofilm into small clusters after 8 h of treatment. According to the real-time quantitative PCR detection results, luteolin could significantly inhibit the relative expression of the biofilm-associated genes *luxS*, *plo*, *rbsB* and *lsrB*. In addition, the in vivo anti-biofilm activity of luteolin against *T. pyogenes* was studied using a rat endometritis model established by glacial acetic acid stimulation and *T. pyogenes* intrauterine infusion. Our study showed that luteolin could significantly reduce the symptoms of rat endometritis. These data may provide new opinions on the clinical treatment of luteolin and other flavonoid compounds on *T. pyogenes* biofilm-associated infections.

## 1. Introduction

*Trueperella pyogenes* (*T. pyogenes*) is often found on the skin surface and mucous membranes of healthy animals’ upper respiratory and urogenital tract. However, it can also develop into pathogenic bacteria in a certain state and infect livestock, wild animals, companion animals and human beings, resulting in suppurative infection of tissue and organ mucosa [1]. As a pathogen, *T. pyogenes* often colonizes the skin, oropharynx, respiratory tract, urogenital tract, gastrointestinal tract and other tissues and organs of domestic animals such as cattle, sheep and pigs, leading to multiple organ inflammation in livestock [2,3].

For cattle, the infection with *T. pyogenes* can cause hysteritis, endometritis, mastitis, liver abscess, suppurative arthritis and pneumonia [4,5,6,7,8]. Endometritis in dairy cattle is a common disease in veterinary clinics, with symptoms such as increased foetal spacing, decreased pregnancy rate, repeat breeding and even being eliminated in severe cases [9]. During calving, the opening of the cattle uterus may cause pathogenic bacterium such as *E. coli*, *S. aureus*, *S. pyogenes* and *T. pyogenes*, etc. to enter the uterus through the birth canal and lead to endometritis. Among various types of those bacteria, *T. pyogenes* was found to have a strong positive correlation with endometritis in dairy cows [10]. By analyzing the dynamics of uterine microbiota in dairy cow uterus contents, *T. pyogenes* was identified as the main species causing clinical endometritis in dairy cows [11]. For example, through screening 1386 dairy cows for clinical endometritis in Switzerland, Ernstberger and colleagues found 28% of the cows had been diagnosed with endometritis, and in these, one of the most predominant microorganisms was the gram-negative bacterium *T. pyogenes*, found in 29.7% of the affected cows [12]. 

Several studies have shown that pathogens, including *T. pyogenes*, could invade host cells and become resistant to antibiotics by forming biofilms [13,14]. About 80% of chronic infections in animals are associated with biofilm bacteria [15]. Biofilm-residing bacteria can even avoid the attack from the host immune system and lead to chronic infection and recurrent infection [16]. The utilization of antibiotics to treat infections caused by biofilms is not an ideal therapeutic solution due to their limited ability to inhibit the formation of biofilm bacteria, remove the mature biofilm bacteria and combat the persister cells in biofilm [17,18,19,20]. 

Chinese herbal medicine was considered a promising drug resource to treat the infections caused by biofilm in recent years due to its special antibacterial mechanism and little drug resistance. Flavonoids that showed excellent antibacterial activity against various resistant strains were found in a wide variety of Chinese herbs such as *Reseda odorata*, *Scutellaria baicalensis Georg*, *Dendranthema morifolium* and *Sophora moorcroftiana*, etc. [21]. Among those compounds, luteolin could inhibit the growth of *T. pyogenes* by influencing the integrity of the cell wall and cell membrane as well as the synthesis of nucleic acid and protein [22]. Luteolin can also increase the susceptibility of antimicrobial-resistant bacteria to antibiotics [23,24]. Moreover, luteolin performs a potent anti-biofilm function. The biofilm formation ability of *E. coli*, *C. albicans* and *E. faecalis* was significantly reduced after being treated with luteolin [25,26]. Our recent research indicated that luteolin could inhibit the ability of TatD DNases which are involved in the biofilm formation of *T. pyogenes* [27].

Herein, we report that when the *T. pyogenes* biofilms were treated with luteolin, their structure was modified, which was investigated and observed by confocal laser microscope and scanning electron microscope, as well as by studying the real-time quantitative PCR detection monitoring the changes in the various biofilm-related coding gene expressions of *luxS*, *plo*, *rbsB* and *lsrB*. In addition, luteolin also showed a significant anti-inflammatory effect on the rat endometritis model established by *T. pyogenes*.

## 2. Results

### 2.1. Anti-Biofilm Activity of Luteolin on T. pyogenes Biofilm

Previous research has shown that the minimum inhibitory concentrations (MICs) of luteolin against *T. pyogenes* isolates were in the range of 39–78 μg/mL, indicating that luteolin has a potent antimicrobial activity on planktonic *T. pyogenes* [22,23]. In this research, we further investigated the anti-biofilm activity of luteolin by measuring its minimum biofilm inhibitory concentrations (MBICs) and minimum biofilm eradication concentration against *T. pyogenes*. Our results showed that luteolin could suppress the formation ability of *T. pyogenes* biofilm and eliminate the maturely formed biofilm, with MBICs in the range of 78–156 μg/mL and the MBECs in the range of 156–312 μg/mL (Table 1).

The confocal laser scanning microscopy (CLSM) and scanning microscopy (SEM) analyses confirmed that there were differences in the *T. pyogenes* BMH06-3 biofilm formation in the presence of 39, 78 and 156 μg/mL of luteolin, in comparison to 0.25 ng/mL ceftiofur. Luteolin was tested at 1/2×, 1×, and 2 × MIC, whereas ceftiofur was tested at 1/2 × MIC. There was no significant difference between the blank control and solvent control groups (Figure 1b and Figure 2b). In the ceftiofur group, the number of killed bacteria in the biofilm were increased, and the depth of the biofilm was reduced significantly (Figure 1c and Figure 2c). However, in the presence of ceftiofur, the biofilm was still formed with a dense structure, indicating that ceftiofur was highly active in inhibiting *T. pyogenes* growth in its planktonic form but was less active against *T. pyogenes* in the biofilm state. The CLSM images exhibited that 39 μg/mL of luteolin had no significant effect on the formation of biofilm, but the SEM images showed a slight influence on the bacterial quantity (Figure 1d and Figure 2d). In contrast, the treatment with 78 μg/mL of luteolin resulted in a moderate proportion of killed bacteria in the *T. pyogenes* biofilm (Figure 2e). After exposure to 156 μg/mL of luteolin, almost no *T. pyogenes* biofilm structure could be observed, cells became scattered and only a few living bacteria remained (Figure 2f and Figure 1f). 

The CLSM and SEM analyses were carried out to visualize the elimination effect of 312 μg/mL (4 × MIC) luteolin on 72 h *T. pyogenes* BMH06-3 biofilms. As shown in Figure 3c and Figure 4c, after exposure to ceftiofur (0.25 ng/mL) for 24 h the depth of biofilms became thinner, and a significant portion of the cells within the biofilms had been killed. However, the remaining bacteria still could form biofilms, indicating that ceftiofur could not completely remove the mature biofilms. Luteolin treatment of 72 h *T. pyogenes* biofilms was found time-dependent where the biofilms were treated with 312 μg/mL luteolin for 2, 4, 6, 8, 10, 12, 16, 20 and 24 h (Figure 3d–l). The biofilm thickness decreased, but no significant change was observed in the bacterial number after the 4 h treatment with luteolin (Figure 3d,e). The extension of the luteolin exposure (8 h) resulted in the dispersion of *T. pyogenes* biofilms. In addition, the depth of the biofilms was reduced to 1/3 of that of the blank group, revealing that luteolin could disperse large pieces of biofilms into small clusters after 8 h treatment (Figure 3f). The exposure of the biofilms with 312 μg/mL luteolin for 12 h caused greater biofilm disruption and a larger proportion of bacterial cell death (Figure 3h and Figure 4d). Almost no biofilm structure could be observed after treatment for 24 h, indicating that 312 μg/mL luteolin could completely disrupt and eliminate the mature *T. pyogenes* biofilms with prolonged treatment (Figure 3l and Figure 4f).

### 2.2. Effect of Luteolin on Biofilm-Associated Gene Expression of T. pyogenes

In order to assess the effect of 1/2 × MBIC luteolin on the biofilm-associated genes of *T. pyogenes* isolates, quantitative real-time PCR (qPCR) analysis was performed. The results revealed different degrees of downregulation of the expression of *luxS* (coding S-ribosylhomocysteine, related to bacterial quorum sensing), *plo* (coding Pyolysin O, pore-forming toxin), *rbsB* (Ribose import binding protein RbsB) and *lsrB* (Autoinducer 2-binding protein LsrB) (Figure 5). Compared with the solvent control group, the expression of *plo* and *lsrB* were down-regulated significantly to ~0.02–0.6-fold when exposed to luteolin. The relative expression of the *plo* gene in HC-H01-4 and BMH05-5 even decreased below 5%. The expression of *luxS* and *rbsB* genes in RY11-3 and HC-H13-2, respectively, were the lowest when treated with luteolin.

### 2.3. In Vivo Treatment of Luteolin on Rat Infectious of T. pyogenes (BMH06-3) Biofilm-Associated Endometritis

The *T. pyogenes* cells were inoculated into the rat uterus and the biofilm was formed on the lens in this model. To assess the rat model, the clinical symptoms of the rats were first studied. The endometritis model rats which developed a huddled and hunched posture showed inactivity, appetite and weight loss, messy and dull hair, and inflamed and moist vulva. The body temperature was significantly increased after the bacterial inoculation. Compared with the endometritis group (0.6491 ± 0.12), the uterine indexes (weight of uterus/body weight of rat × 100) of the ceftiofur group (0.3665 ± 0.09 **), luteolin H (0.1491 ± 0.03 **) and M (0.1484 ± 0.03 **) groups were significantly decreased. 

The bacteria of the intrauterine fluid were isolated and cultured. The *plo* gene screening results showed that no pathogen was contained in the blank control group. In contrast, a large number of *T. pyogenes* was found in the model group (Appendix A). The CLSM examination result illustrated that a dense biofilm was formed maturely on the silicone hydrogel lenses of the control group (Figure 6a). After the treatment with ceftiofur, the number of dead bacteria increased significantly but the biofilm still formed thickly (Figure 6b). After the in vivo treatment with luteolin, the biofilms of the M and H luteolin groups were found loose and porous, but could not be disrupted completely (Figure 6d,e).

The rats in the treatment groups (M and H groups) showed weight gain after the luteolin administration. Table 2 presents the data on leukocyte formation before and following treatment in various groups. The total leukocyte count and neutrophil count were increased while the lymphocyte count was decreased significantly in the endometritis group (E), which indicates that the bacterial infection was serious after the *T. pyogenes* perfusion. After the treatment with a high concentration of luteolin (150 mg/kg b.w), the leukocyte, lymphocyte, monocyte and neutrophil count were all within the reference range, indicating that the bacterial infections of the rats in the H group had moved to the decline stage.

The anatomical form of the rat uterus in the control group did not adhere to the surrounding-tissue and was uniform in texture (Figure 7a). In the endometritis group, organs in the pelvis were adhesive and congested, and the uterus volume was greatly increased with obvious swelling and convoluted shape. The uterine wall was thinning, and the uterine lumen was filled with yellow pus (Figure 7b). The swelling of the uterus, as well as endometrial hyperemia and edema, could also be observed after the treatment of the low concentration of luteolin (50 mg/kg b.w) (Figure 7d and Figure 8d). The histopathologic analysis shows that the uterus of rats in the ceftiofur group presents vascular dilation and congestion phenomenon with a little inflammatory cell infiltration (Figure 8c). However, the uteruses of rats in the luteolin H and M groups only showed slight hyperemia and inflammatory cell infiltration, but the morphology was normal without obvious lesions (Figure 7e,f and Figure 8e,f). 

## 3. Discussion

Most clinical pathogenic bacteria biofilm can form on the surface of various systems such as internal environment systems, synthetic materials, contact lenses, etc. [28]. The formation of biofilms also increases virulence and drug resistance, resulting in chronic or persistent infections in animals, and thus increasing the difficulty of treatment [29]. Most antimicrobial agents preferentially interact with the substrate of biofilm bacteria, leading to a significant increase in drug resistance [30]. The development of naturally derived antimicrobial agents to inhibit the formation of bacterial biofilm is expected to improve the effectiveness of clinical treatments or bacterial infectious diseases.

Luteolin as the representative of a kind of flavonoids, has outstanding anti-inflammatory, antibacterial, anti-tumor, anti-allergic and other pharmacological activities [22,31]. The anti-biofilm results in this study showed that the MBIC and MBEC values of Luteolin against *Trueperella pyogenes* (*T. pyogenes*) isolates were 78 or 156 µg/mL (2 × MIC) and 156 or 312 µg/mL (4 × MIC), respectively. After treatment with 156 μg/mL luteolin for 72 h, *T. pyogenes* could scarcely form biofilm. An amount of 312 μg/mL of luteolin could even eradicate the mature biofilm. However, for general antibiotics, the resistance of biofilm is 10–1000 times that of planktonic bacteria [32]. Compared with antibiotics, the potent anti-biofilm performance of luteolin is probably due to its multi-target, anti-bacterial mechanism against *T. pyogenes* [22,23,24]. 

The transcriptional levels of biofilm formation-related genes (*luxS*, *plo*, *rbsB* and *lsrB*) of *T. pyogenes*, with or without the treatment of luteolin, were analyzed to study the anti-biofilm mechanism of luteolin. The biofilm formation of *T. pyogenes* is partly dominated by the PloS/PlosR two-component regulatory system [33]. The qPCR result in this study showed that the *plo* gene-relative transcript level decreased significantly after luteolin treatment. Pyolysin (PLO), which belongs to a cholesterol-dependent cytolysin family, is considered the major virulence factor and a host-protective antigen of *T. pyogenes* [8]. Therefore, luteolin probably could reduce the virulence and inhibit the biofilm formation of *T. pyogenes* simultaneously by influencing the expression of *plo*, thus exerting its antimicrobial therapeutic effect. Furthermore, the mRNA transcription level of the *luxS*, *rbsB* and *lsrB* genes, which are related to the quorum-sensing system luxS/AI-2, was decreased to a certain extent after luteolin treatment [34,35]. The biofilm formation capacity of the Δ*luxS* strain was significantly weakened and the virulence and pathogenicity of the strain were also affected [36]. It is speculated that luteolin may affect the normal function of the LuxS/AI-2 Quorum sensing system of *T. pyogenes* by inhibiting the expression of *luxS*, *rbsB* and *lsrB* genes and giving play to anti-biofilm effects. 

Endometritis was reported to partly result from a *T. pyogenes* infection of the endometrium in dairy cows [37]. In this study, a *T. pyogenes* isolate (BMH06-3), which was isolated from a cow with endometritis, was used to establish the endometritis model [27]. The uterine perfusion of the bacterial suspension will probably cause metritis or vaginitis instead of endometritis in a rat model. Therefore, in order to establish the rat endometritis model, 3% acetic acid was injected into uterus horns for 3 days to impair the uterus chemically prior to the bacterial perfusion in this study [38]. The injection of estradiol benzoate into the irritable parous rat was aimed at improving their tractability, which is necessary for further operation procedures. The animal experiment results indicated that luteolin shows a good therapeutic effect on *T. pyogenes* biofilm-associated rat endometritis.

Comparing the endometritis groups, the total leukocyte counts of the luteolin high and medium groups were significantly decreased to the standard range, demonstrating that the endometritis of rats induced by *T. pyogenes* was lightened by luteolin. However, the leukocyte populations of rats in the ceftiofur group had not returned to the reference level. The uterus morphology and histopathologic examination results showed that although the rat uterus of ceftiofur groups returned to normal size, the extravasated blood was still present in the rat uterus horn, whereas rats had improved conditions after the treatment with luteolin. Compared with ceftiofur, luteolin had a better therapeutic effect on *T. pyogenes* biofilm-associated endometritis, and possibly not only due to its antimicrobial and anti-biofilm effects. Luteolin could decrease the expression level of proinflammatory cytokines (TNF-a, IL-1b, and IL-6) and increase the level of IL-38 in rat inflammation models [39,40]. Therefore, the significant effect of luteolin treatment in this study may also be related to its extraordinary anti-inflammation function.

In summary, these results indicate that Luteolin is a promising anti-biofilm agent against *T. pyogenes*. Luteolin could significantly inhibit the expression of the biofilm-related coding genes *luxS*, *plo*, *rbsB* and *lsrB*. Luteolin also showed a potent therapeutic effect on the *T. pyogenes* biofilm-associated rat endometritis model. This study has provided new insight into the possible clinical application of naturally derived flavonoid compounds as well as their roles as potential candidates for anti-biofilm drug development.

## 4. Materials and Methods

### 4.1. Drug and Bacterial Strain Information

Luteolin with 98% purity was purchased from PureOne Biotechnology (Shanghai, China) and dissolved in 1% dimethyl sulfoxide (DMSO, Sigma–Aldrich, Shanghai, China) to produce a stock solution. Ceftiofur with 92.1% purity was purchased from Ryon biological technology (Shanghai, China). *T. pyogenes* isolates (Table 3) were collected from dairy cows with endometritis in Inner Mongolia, China which was mentioned previously [41]. The *T. pyogenes* strain (ATCC 19411) was selected as the reference strain. *T. pyogenes* strains were inoculated on Mueller-Hinton agar (MHA, Solarbio, Beijing, China) containing 5% (*v*/*v*) defibrinated sheep blood (Hopebio, Qingdao, China) and cultured in brain–heart infusion (BHI) medium (Solarbio, Beijing, China) supplemented with 8% (*v*/*v*) fetal bovine serum (FBS, Gibco, Grand Island, NE, USA).

### 4.2. Biofilm Susceptibility Test

For measuring the minimum biofilm inhibitory concentration (MBIC), the luteolin solutions from 19 to 5000 µg/mL were two-fold diluted were incubated with 10^5^ CFU/mL of bacterial culture and placed in the wells of a 96-well plate (200 μL per well). After 24 h of incubation (at 37 °C and 200 rpm), when the biofilm was formed, each well was rinsed twice with PBS (pH 7.4, Solarbio, Beijing, China) to remove the planktonic bacterium. The attached biofilm in each well was scrapped and suspended in 250 μL of PBS. From each well, 10 μL of a mixture was inoculated on MHA plates with 5% blood. After being cultured at 37 °C for 24 h, the colony-forming units (CFU) of the test groups were compared with the medium control group. The lowest corresponding luteolin concentration, in which the recorded CFU is less than or equal to the solvent control group (0.01% DMSO), was defined as the minimum biofilm inhibitory concentration (MBIC). 

For measuring the minimum biofilm eradication concentration (MBEC), a similar procedure was followed except that 200 μL of each bacterial culture was first seeded in a 96-well plate and incubated for 24 h (at 37 °C and 200 rpm) to form the biofilm. After the planktonic phase was removed, luteolin solutions in the standard concentrations were added to the biofilms that had formed in each well. After 24 h incubation, the biofilms were suspended and the CFU was recorded as described above for MBIC.

### 4.3. Biofilm Imaging

The intervention effects of luteolin on biofilm were observed by confocal laser scanning microscopy (CLSM) and scanning electron microscopy (SEM). For CLSM examination, the bacterial suspension (10^5^ CFU/mL) was incubated in glass-bottom cell culture dishes (NEST, Wuxi, China) for 24 h. After rinsing twice with PBS (pH 7.2) to remove the planktonic bacterium, Syto-9—which stains DNA in all cells—and propidium iodide (PI)—which stains DNA only in dead cells (Invitrogen, Camarillo, CA, USA)—were incubated with the samples for 15 min according to the LIVE/DEAD BacLight Bacterial Viability Kit protocol. Images were captured with a confocal microscope (Leica Camera AG, Solms, Germany). 

For the SEM examination, the bacterial suspension (10^5^ CFU/mL) was incubated on cell slides in the 24-well plate for 72 h to form the mature biofilms. The cell slides were rinsed three times with PBS (pH 7.2), and further fixation was performed with 2.5% glutaraldehyde (Sigma, Shanghai, China) overnight at 4 °C at room temperature and then rinsed three times with PBS. Then, the sample was dehydrated in an ethanol gradient (50%, 70% and 90%, each for 10 min) followed by 100% ethanol twice for 10 min each time and finally dried. After platinum coating, samples were imaged using an electron microscope (HITACHI, Tokyo, Japan). 

For both two microscopic visualizations, to observe the influence of luteolin on the formation of biofilm, the drug solutions (with final concentration at 8MIC, 4MIC and 2MIC) were added to the culture dishes or well plates together with the bacterial suspension. To observe the elimination of luteolin on biofilm, the luteolin solutions were added to the biofilms that had formed, followed by incubating for 2, 4, 6, 12, 18 and 24 h to obtain samples at different time points. DMSO (0.01%) and ceftiofur (0.25 μg/mL) served as the solvent control and positive medicine control.

### 4.4. Biofilm-Associated Gene Expression Analysis

Bacteria collected from biofilm with/without luteolin treatment were centrifuged for 10 min at 5000× *g*. Total RNAs from the biofilm were extracted and purified using the Trizol reagent as described by the manufacturer (Invitrogen, Carlsbad, CA, USA). The quality and integrity of RNAs were checked as described previously [24]. The cDNA of the total RNA was synthesized by using TransScript One-Step gDNA Removal and cDNA Synthesis SuperMix (Transgen, Beijing, China). The RT-PCR was performed using a TB Green^®^ Premix Ex Taq™ II Kit (TaKaRa, Dalian, China) on an Applied Biosystems^®^ QuantStudio^®^ 3 Real-Time PCR System (Thermo Fisher, Waltham, MA, USA) according to the manufacturer’s instructions. The *gapA* of *T. pyogenes* in biofilm was used as a reference gene for comparison in qRT–PCR. The primer sequences are shown in Table 4. The qRT–PCR cycling conditions were as follows: 95 °C for 30 s and 45 cycles of 5 s at 95 °C, 30 s at 60 °C, and 30 s at 72 °C. The relative expression of the biofilm-related genes *luxS*, *plo*, *rbsB* and *lsrB* were normalized against that of *gapA* and quantified using the 2^−ΔΔCt^ method.

### 4.5. Animal Experiment

To establish the rat *T. pyogenes* biofilm-associated endometritis model, female parous Sprague-Dawley (SD) rats (16–22 weeks of age) were purchased from Changsheng Biological Technology Co., Ltd. (Shenyang, China, Permit No.: SCXK (Liao) 2020-0001) and maintained under specific pathogen-free conditions for experiments. Before the bacteria infusion, the rats were intramuscularly injected with estradiol benzoate (0.1 mg/kg/day) for 3 days to cause artificial estrus [38]. For stimulation, 3% acetic acid was injected into uterus horns for 3 days. On day three, after the acetic acid stimulation, the rat uterine was inoculated with 2 mL bacterial suspension containing 1.0 × 10^9^ CFU/mL of *T. pyogenes* (BMH06-3) by uterine perfusion daily for 3 days [42] A silicone hydrogel lens (Hydron, Shanghai, China) was implanted into the rat uterus after the first time of bacterial inoculation to form the biofilm [43]. After the induction of the endometritis model, the biofilm formed on the silicon hydrogel lenses was observed by CLSM to show its existence in the rat uterus. Meanwhile, the clinical symptoms, the body temperature, the white blood cell classification alterations, the bacterial content inside the uterus model, and the morphology and pathological alterations of the rat were observed to determine whether the model was induced successfully or not.

For the drug treatment, animals were randomly divided into six groups (*n* = 7): the control group (C), endometritis group (E), ceftiofur group (CE) and three different luteolin treatment groups (L, M and H). The control and endometritis groups were intraperitoneally given NaCl 0.9%. The ceftiofur group was treated with ceftiofur (1.5 mg/kg b.w) in an attempt to treat the uterine infection. Luteolin treatment groups were treated with luteolin (L, 50 mg/kg b.w; M, 100 mg/kg b.w; and H, 150 mg/kg b.w, respectively). All administrations were performed continuously for 3 days. Seventy-two hours after the last treatment, rats were killed by cervical dislocation after chloral hydrate anesthesia. The uterine tissue was isolated and the uterine index (uterine mass/animal mass × 100%) was calculated. About 2 cm in the middle of the uterine horn was cut and preserved with 4% paraformaldehyde (Dingguo, Beijing, China) and made into pathological sections. The intrauterine implanted silicon hydrogel was removed, stained with SYTO9/PI and observed with CLSM (Leica Camera AG, Solms, Germany). Meanwhile, blood samples were collected from rat tail veins and the routine tests were performed by an automatic hematology analyzer (HF-3000, Hanfang, Jinan, China). All animal procedures performed in the present study were conducted according to the animal husbandry guidelines of Shenyang Agricultural University. The Ethics Committee of Shenyang Agricultural University approved the laboratory animal experiments [Permit No.: SYXK (Liao) 2021-0010].

## Figures and Tables

**Figure 1 ijms-23-14451-f001:**
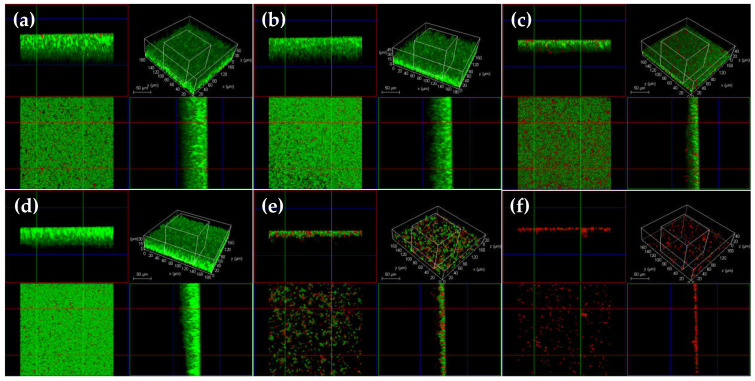
CLSM analysis of *T. pyogenes* isolate (BMH06-3) biofilm co-incubated with luteolin for 72 h (Green: live bacteria; red: killed bacteria; black: no bacteria). (**a**) Blank control biofilm. (**b**) Solvent control biofilm (1% DMSO). (**c**) Biofilm co-incubated with ceftiofur (0.25 ng/mL). (**d**–**f**) Biofilm co-incubated with luteolin (39, 78 and 156 μg/mL).

**Figure 2 ijms-23-14451-f002:**
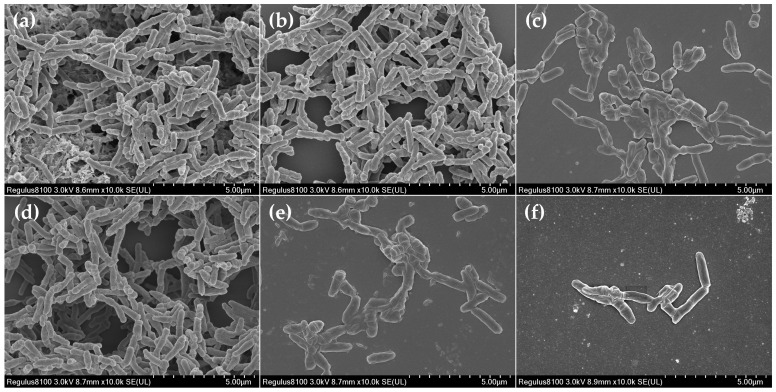
SEM analysis of *T. pyogenes* (BMH06-3) biofilm co-incubated with luteolin for 72 h. (**a**) Blank control biofilm. (**b**) Solvent control biofilm (1% DMSO). (**c**) Biofilm co-incubated with ceftiofur (0.25 ng/mL). (**d**–**f**) Biofilm co-incubated with luteolin (39, 78 and 156 μg/mL).

**Figure 3 ijms-23-14451-f003:**
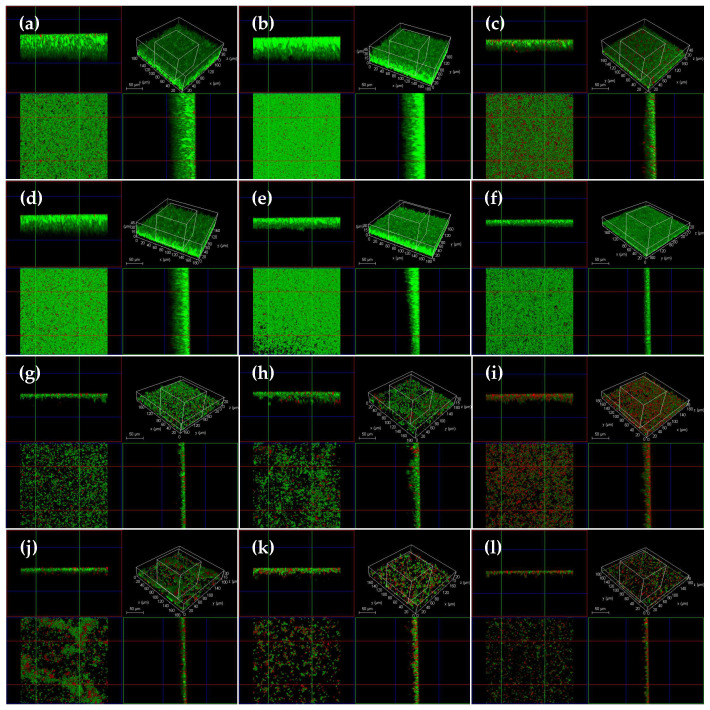
CLSM analysis of mature *T. pyogenes* (BMH06-3) biofilm disruption treated with 312 μg/mL luteolin (Green: live bacteria; red: killed bacteria; black: no bacteria). (**a**) Blank control biofilm. (**b**) Solvent control biofilm (1% DMSO). (**c**) Mature biofilm treated with ceftiofur (0.25 ng/mL) for 24 h. (**d**–**l**) Mature biofilm treated with 312 μg/mL luteolin for 2, 4, 6, 8, 10, 12, 16, 20 and 24 h.

**Figure 4 ijms-23-14451-f004:**
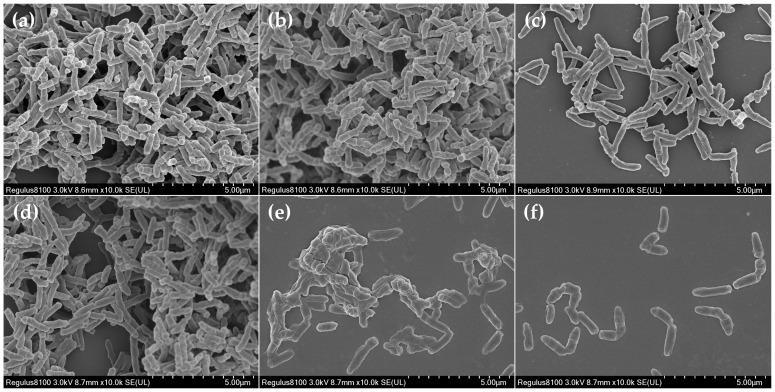
SEM analysis of mature *T. pyogenes* (BMH06-3) biofilm disruption treated with 312 μg/mL luteolin. (**a**) Blank control biofilm. (**b**) Solvent control biofilm (1% DMSO). (**c**) Mature biofilm treated with ceftiofur (0.25 ng/mL) for 24 h. (**d**–**f**) Mature biofilm treated with 312 μg/mL luteolin for 12, 18 and 24 h.

**Figure 5 ijms-23-14451-f005:**
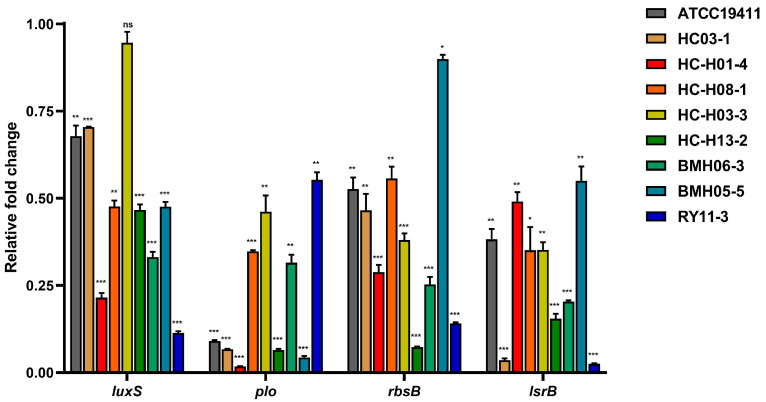
Quantitative real-time RT-PCR analysis of genes involved in biofilm formation of *T. pyogenes* isolates. Biofilms were formed in the presence of 1/2 MBIC luteolin. Housekeeping genes *gapA* were used for normalization. Error bars and asterisks indicate the SEM. statistical significance of difference was analysed by *t*-test (* *p* < 0.05; ** *p* <0.01; *** *p* < 0.001).

**Figure 6 ijms-23-14451-f006:**
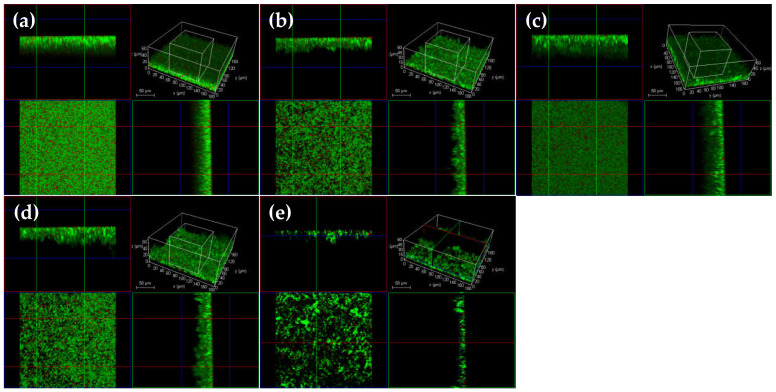
CLSM analysis of *T. pyogenes* (BMH06-3) biofilm on the silicon hydrogel lenses after the in vivo treatment of luteolin (Green: live bacteria; red: killed bacteria; black: no bacteria). (**a**) Blank control biofilm. (**b**) Ceftiofur group (1.5 mg/kg b.w). (**c**–**e**) Luteolin group (50 mg/kg b.w; 100 mg/kg b.w; 150 mg/kg b.w).

**Figure 7 ijms-23-14451-f007:**
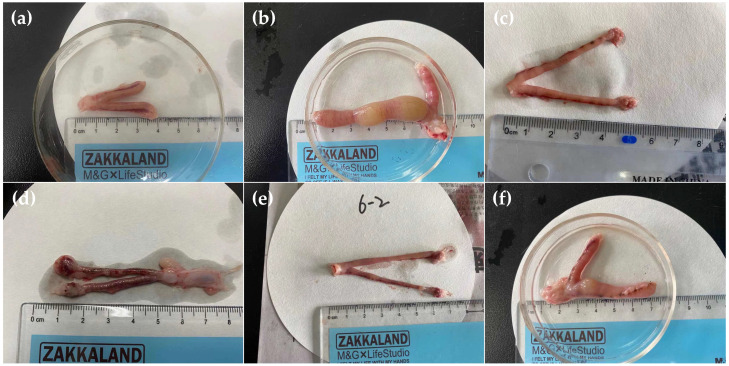
Macroscopic uterus morphology comparison of rats in various groups. (**a**) Control group. (**b**) Endometritis group. (**c**) Ceftiofur group. (**d**–**f**) Luteolin treatment groups (L, 50 mg/kg b.w; M, 100 mg/kg b.w; H, 150 mg/kg b.w, respectively).

**Figure 8 ijms-23-14451-f008:**
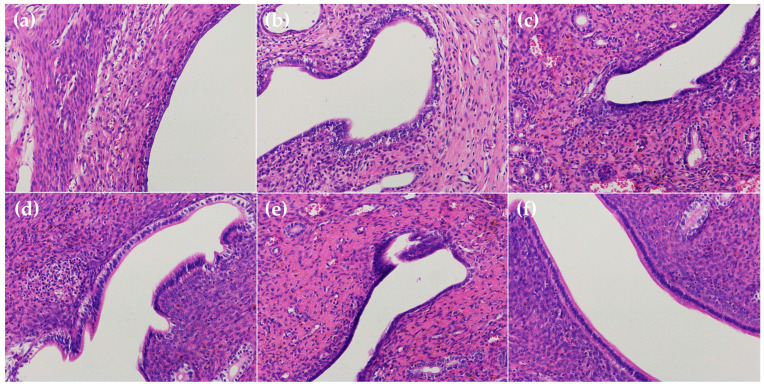
Histopathologic examinations of H & E stained uterine of rats in various groups (200×). (**a**) Control group. (**b**) Endometritis group. (**c**) Ceftiofur group. (**d**–**f**) Luteolin treatment groups (L, 50 mg/kg b.w; M, 100 mg/kg b.w; H, 150 mg/kg b.w, respectively).

**Table 1 ijms-23-14451-t001:** Antimicrobial activity and anti-biofilm activity of luteolin against type stain and clinical isolates.

*T. pyogenes*	MIC (μg/mL)	MBIC (μg/mL)	MBEC (μg/mL)
ATCC19411	78	78	156
HC03-1	78	78	156
HC-H01-4	78	156	312
HC-H08-1	78	156	312
HC-H03-3	78	156	312
HC-H13-2	78	156	312
BMH06-3	78	156	312
BMH05-5	39	156	312
RY11-3	78	156	156

**Table 2 ijms-23-14451-t002:** Mean ± SEM values of leukocyte count in various experimental groups.

Groups	Total Leukocyte (10^9^/L)	Lymphocyte (%)	Monocyte (%)	Neutrophil (%)
Reference range	8.93–6.42	63.16–38.36	2.55–0.00	53.30–32.59
Control (C)	7.35 ± 1.97	56.24 ± 6.21	1.12 ± 0.08	34.65 ± 1.57
Endometritis (E)	11.52 ± 4.54	35.67 ± 2.15	2.61 ± 0.97	59.31 ± 9.46
Ceftiofur (CE)	8.62 ± 2.71	46.24 ± 2.18	1.91 ± 0.15	41.26 ± 3.21
Luteolin (L)	7.98 ± 2.16	49.21 ± 6.21	2.01 ± 0.61	43.26 ± 4.67
Luteolin (M)	7.61 ± 4.51	57.21 ± 4.79	1.32 ± 0.12	31.58 ± 2.37
Luteolin (H)	9.87 ± 3.14	39.46 ± 4.12	1.16 ± 0.24	53.18 ± 2.18

**Table 3 ijms-23-14451-t003:** Source of *T. pyogenes* isolates.

*T. pyogenes* Isolates	Source
HC03-1	Endometritis
HC-H01-4	Endometritis
HC-H08-1	Endometritis
HC-H03-3	Endometritis
HC-H13-2	Endometritis
BMH06-3	Endometritis
BMH05-5	Endometritis
RY11-3	Endometritis
HC03-1	Endometritis

**Table 4 ijms-23-14451-t004:** Primers used for RT-PCR.

Gene		Primer Sequence
*luxS*	F	ACGCAACCACGCCGATAACG
	R	CGCAAGGTGACCTCGATCAGTTC
*plo*	F	TTGCCTCCAGTTGACGCTTTGAC
	R	GCCTTCTCGACGGTTGGATTCAG
*rbsB*	F	TCGTGTTTGCCGCTTTGGAC
	R	AACCGTGGTGACCGGAATGT
*lsrB*	F	GGTTCTCAACGCCAAATGCTATGAAG
	R	CGCCGACCTTGATAGATTTACCAGAG
*gapA*	F	CGGCGAAGAACGAGGACATCAC
	R	GTCGGCAGTGTAGGCGTGAAC

## Data Availability

The data presented in this study are available in the article/Appendix A, further inquiries can be directed to the corresponding authors.

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
