# Peer review of "Effects of Luteolin on Biofilm of Trueperella pyogenes and Its Therapeutic Effect on Rat Endometritis"

_ijms, 2022, doi:10.3390/ijms232214451_

Round 1
Reviewer 1 Report
The Authors submitted the Ms “Effects of luteolin on biofilm of Trueperella pyogenes and its therapeutic effect on rat endometritis. Luyao Zhang, Yitong Cai, Lishuang Li, Chen Chen, Hanyu Zhao, Zehui Zhang, Yaochuan Liu, Yingyu Wang, Chunlian Tian and Mingchun Liu”. They informed that Trueperella pyogenes (T. pyogenes) often exist on the skin surface and mucous membranes of the upper respiratory tract and urogenital tract of healthy animals. And they can develop into pathogenic bacteria in a certain state and infect livestock, wild animals, companion animals and human beings, resulting in suppurative infection of tissue and organ mucosa. As a pathogen, T. pyogenes colonize the skin, oropharynx, respiratory tract, urogenital tract, gastrointestinal tract and other tissues and organs of domestic animals such as cattle, sheep and pigs, leading to multiple organ inflammation in livestock. For cattle infection with T. pyogenes can cause hysteritis, endometritis, mastitis, liver abscess, suppurative arthritis and pneumonia. Endometritis in dairy cattle is a common disease in the veterinary clinic, with symptoms such as increased foetal spacing, decreased pregnancy rate, repeat breeding. During calving, the opening of the cattle uterus may cause pathogenic bacterium such as E. coli, S. aureus, S. pyogenes and T. pyogenes, etc. to enter the uterus through the birth canal and lead to endometritis. The bacteria, T. pyogenes have a strong positive correlation with endometritis in dairy cows. T. pyogenes was identified as the main species causing clinical endometritis in dairy cows. In Switzerland 1386 dairy cows were investigated, ~ 28% of them suffered from endometritis and 29.7% was T. pyogenes. The formation of biofilms also increases virulence and drug resistance, resulting in chronic or persistent infections in animals, and thus increasing the difficulty of treatment. When T. pyogenes invade host cells, they become resistant to antibiotics by forming biofilms. ~ 80% of chronic animal’s infections are biofilm bacteria. Biofilm-residing bacteria can avoid the attack from the host immune system and lead to chronic infection and recurrent infection.
The development of new anti-biofilm drugs will improve the current status of clinical treatment for T. pyogenes-induced infections. Luteolin is a natural-derived flavonoid compound with antibacterial activities. In this study, the effects and the mechanism of luteolin action on T. pyogenes biofilm were investigated. Luteolin inhibited the growth of T. pyogenes by influencing the integrity of the cell wall and cell membrane as well as the synthesis of nucleic acid and protein. It can increase the susceptibility of antimicrobial-resistant bacteria to antibiotics.
- I suggest re-writing the phrase “Herein, the alterations of T. pyogenes biofilm which was treated with luteolin were analyzed and observed. The anti-biofilm mechanism of luteolin was explored using monitoring the change of various biofilm-related coding gene expressions of T. pyogenes which was influenced by luteolin. To further understand the therapeutic function of luteolin on infectious caused by the biofilm of T. pyogenes, a rat model of endometritis was developed”
- By - When the T. pyogenes biofilms were treated with luteolin, its structure was modified, which were investigated and observed by confocal laser microscope and scanning electron microscope, as well as studying the real-time quantitative PCR detection monitoring the change of various biofilm-related coding gene expressions of luxS, plo, rbsB and lsrB. And its effect on rat endometritis model.
Results. - Luteolin (312 μg/mL), a high concentration was able to suppress the formation ability of T. pyogenes biofilm and eliminated the mature formed biofilm, after 24 hour of treatment. The effect of 1/2 MBIC luteolin on biofilm-associated gene expression of T. pyogenes was tested analyzed by quantitative real-time PCR (qPCR). The results revealed different degrees of down regulation of the expression of luxS, plo, rbsB and lsrB. The authors may suggest that luteolin affect the normal function of LuxS/AI- 240 2 Quorum sensing system of T. pyogenes by inhibiting the expression of luxS, rbsB and lsrB genes and gives play to anti-biofilm effects. The in vivo rat infected with T. pyogenes (BMH06-3) biofilm-associated endometritis were treated with luteolin, the M and H rats groups shows a weight gain after the luteolin administration. The total leukocyte and neutrophil count increased while the lymphocyte count decreased significantly in the endometritis group (E) which indicates that the bacterial infection is serious after the T. pyogenes perfusion, but after the treatment with 150 mg/kg luteolin, the leukocyte, lymphocyte, monocyte and neutrophil count were all within the reference range, indicating that the bacterial infection of rats in the H group had moved to the decline stage. The anatomical form of the rat uterus in the control group was no adherence with surround-tissue and with uniform texture. In the endometritis group, organs in the pelvic were adhesive and congested, the uterus volume was greatly increased and with obvious swelling and convoluted shape. The uterine wall is thinning, and the uterine lumen is filled with yellow pus. However, the uteruses of rats in luteolin H and M groups only show slight hyperemia and inflammatory cell infiltration, but the morphology was normal without obvious lesions. Luteolin decreased the expression level of proinflammatory cytokines (TNF-a, IL-1b, and IL-6) and increased the level of IL-38 in inflammation model rats.
According with result presented in the Ms, I suggest English grammar correction. The result will be interesting for the Journal and audience. The discussion is good, excellent abstract. Introduction is O.K.
Author Response
Dear reviewer,
The authors would like to thank you for your comments. Care has been taken to improve the work and address concerns as per the specific comments below.
Point 1: I suggest re-writing the phrase “Herein, the alterations of T. pyogenes biofilm which was treated with luteolin were analyzed and observed. The anti-biofilm mechanism of luteolin was explored using monitoring the change of various biofilm-related coding gene expressions of T. pyogenes which was influenced by luteolin. To further understand the therapeutic function of luteolin on infectious caused by the biofilm of T. pyogenes, a rat model of endometritis was developed”
By - When the T. pyogenes biofilms were treated with luteolin, its structure was modified, which were investigated and observed by confocal laser microscope and scanning electron microscope, as well as studying the real-time quantitative PCR detection monitoring the change of various biofilm-related coding gene expressions of luxS, plo, rbsB and lsrB. And its effect on rat endometritis model.
Response 1: Many thanks for your suggestion, we revised the phrase as "Herein, we report that when the T. pyogenes biofilms were treated with luteolin, its structure was modified, which were investigated and observed by confocal laser microscope and scanning electron microscope, as well as studying the real-time quantitative PCR detection monitoring the change of various biofilm-related coding gene expressions of luxS, plo, rbsB and lsrB. In addition, luteolin also showed significant anti-inflammatory effect on the rat endometritis model which established by T. pyogenes biofilm." Line 74-79
Point 2. According with result presented in the Ms, I suggest English grammar correction.
Response 2: Many thanks for your suggestion. We performed following grammar corrections.
(1) We revised "Trueperella pyogenes is a species of conditional pathogen that can cause a variety of suppurative infections in veterinary clinics " as " Trueperella pyogenes is an opportunistic pathogen that causes suppurative infections in animals " Line 13
(2) We revised “improve the current status of clinical treatment for T. pyogenes-induced infections” as “improve the current treatment status of for controlling T. pyogenes infections in animal husbandry industry”. Line 15
(3) We revised “The result indicated that” as “The results indicated that”. Line 19
(4) We revised “The results showed that” as “Our study showed that”. Line 24
(5) We revised “Trueperella pyogenes (T. pyogenes) often exist on the skin surface and mucous membranes of the upper respiratory tract and urogenital tract of healthy animals.” as “Trueperella pyogenes (T. pyogenes) is often found on the skin surface and mucous membranes of healthy animals’ upper respiratory and urogenital tract”. Line 31
(6) We revised “T. pyogenes often colonize the skin” as “T. pyogenes often colonizes the skin”. Line 35
(7) We revised “By analysing the metagenomic dynamic analysis of dairy cow uterus contents” as “By analyzing the dynamics of uterine microbiota in dairy cow uterus contents”. Line 47
(8) We revised “Besides, 1386 dairy cows in Switzerland were investigated, about 28% of them suffered from endometritis and the detection rate of T. pyogenes was 29.7%” as “For example, through screening, 1386 dairy cows for clinical endometritis in Switzer-land, Ernstberger and colleagues identified 28% of the cows diagnosed with endometritis, from which one of the most predominant microorganisms was gram-negative bacterium, T. pyogenes 29.7%”. Line 48-51
(9) We revised “Luteolin performed potency anti-biofilm function” as “luteolin performed a potent anti-biofilm function”. Line 69
(10) We revised “which involved in the biofilm formation” as “which are involved in the biofilm formation”. Line 72
(11) We revised “Based on the previous research, the MICs of luteolin against T. pyogenes isolates ranged from 39 to 78 μg/mL” as “Previous research has shown that, the minimum inhibitory concentrations (MICs) of luteolin against T. pyogenes isolates were in the range of 39 to 78 μg/mL,”. Line 83-85
(12) We added “In this research, we further investigated the anti-biofilm activity of luteolin by measuring its minimum biofilm inhibitory concentrations (MBICs) and minimum biofilm eradication concentration against T. pyogenes.” Line 85-87
(13) We revised “The Biofilm susceptibility test result showed that the MBICs range of T. pyogenes strains was 78 to 156 μg/mL. The MBECs values of 9 strains of test strains were in the range of 156 to 312 μg/mL” as “Our results showed that luteolin could suppress the formation ability of T. pyogenes biofilm and eliminate the mature formed biofilm, with MBICs in the range 78-156 μg/mL and the MBECs in the range 156-312 μg/mL”. Line 89-90
We removed “It can be seen that luteolin was able to suppress the formation ability of T. pyogenes biofilm broadly. Besides, a high concentration of luteolin could eliminate the mature formed biofilm of T. pyogenes” accordingly. Line 90
(14) We revised “The CLSM and SEM analysis of biofilm formation by T. pyogenes BMH06-3 in the presence of 39, 78 and 156 μg/mL of luteolin or 0.25 ng/mL ceftiofur were performed” as “The confocal laser scanning microscopy (CLSM) and scanning microscopy (SEM) analyses confirmed there were differences in the T. pyogenes BMH06-3 biofilm formation in the presence of 39, 78 and 156 μg/mL of luteolin, in comparison of 0.25 ng/mL ceftiofur”. Line 94-96
(15) We revised “Luteolin was tested at 1/2, 1 and 2 × MIC” as “Luteolin was tested at 1/2 ×, 1 × and 2 × MIC”. Line 97
(16) We revised “between the blank control group and the solvent control group” as “between the blank control and solvent control groups”. Line 98
(17) We revised “the number of killed bacteria in biofilm was increased, and the depth of biofilm was reduced significantly” as “the numbers of killed bacteria in the biofilm were increased, and the depth of the biofilm was reduced significantly”. Line 99
(18) We revised “However, the biofilm treated by ceftiofur was still formed with a dense structure which indicated that though ceftiofur had good anti-bactericidal ability it could not inhibit the biofilm form effectively” as “However, in the presence of ceftiofur, the biofilm was still formed with a dense structure, indicating that ceftiofur was highly active in inhibiting T. pyogenes growth in its planktonic form but was less active against T. pyogenes in the biofilm state”. Line 101-103
(19) We revised “CLSM images exhibit that 39 μg/m of luteolin” as “CLSM images exhibited that 39 μg/mL of luteolin”. Line 104
(20) We revised “SEM image shows a slight influence” as “SEM images showed a slight influence”. Line 105
(21) We revised “the treatment of 78 μg/m of luteolin resulted in a moderate proportion of killed biofilm bacteria” as “the treatment of 78 μg/mL of luteolin resulted in a moderate proportion of killed bacteria in the T. pyogenes biofilm”. Line 106
(22) We revised “After the treatment of 156 μg/mL luteolin, almost no biofilm structure could be observed, cells become scattered and only a few living bacteria remain” as “After exposure to 156 μg/mL of luteolin, almost no T. pyogenes biofilm structure could be observed, cells became scattered and only a few living bacteria remained”. Line 107-109
(23) We re-wrote the paragraph “The CLSM and SEM analysis of the elimination effect of 312 μg/mL (4 × MIC) luteolin on mature biofilm…….” Line 119-135
By “The CLSM and SEM analyses were carried out to visualize the elimination effect of 312 μg/mL (4 × MIC) luteolin on 72 h T. pyogenes BMH06-3 biofilms. As shown in Figure 3 c and Figure 4 c, after exposure to ceftiofur (0.25 ng/mL) for 24 h, the depth of biofilms became thinner, and a significant portion of the cells within the biofilms was killed. However, the remained bacteria still could form the biofilms, indicating that ceftiofur could not completely remove the mature biofilms. Luteolin treatment of 72 h T. pyogenes biofilms was found time-dependent where the biofilms were treated with 312 μg/mL luteolin for 2, 4, 6, 8, 10, 12, 16, 20 and 24 h (Figure 3 d-l). The biofilm thickness decreased, but no significant change was observed in the bacterial number after the 4 h treatment of luteolin (Figure 3 d and e). Extension of the luteolin exposure (8 h) resulted in the dispersion of T. pyogenes biofilms. In addition, the depth of the biofilms was reduced to 1/3 of that of the blank group, revealing that luteolin could disperse large pieces of biofilms into small clusters after 8 h treatment (Figure 3 f). Exposure of the biofilms with 312 μg/mL luteolin for 12 h caused greater biofilm disruption and a larger proportion of bacterial cell death (Figure 3 h and Figure 4 d). Almost no biofilm structure could be observed after treatment for 24 h, indicating that 312 μg/mL luteolin could completely disrupt and eliminate the mature T. pyogenes biofilms with prolonged treatment (Figure 3 l and Figure 4 f).” Line 119-135
(24) We revised “In order to assess the effect of 1/2 MBIC luteolin on biofilm-associated genes of T. pyogenes isolates, quantitative real-time PCR (qPCR) analysis was performed, and” as “In order to assess the effect of 1/2 × MBIC luteolin on biofilm-associated genes of T. pyogenes isolates, quantitative real-time PCR (qPCR) analysis was performed”. Line 147
(25) We revised “a large number of T. pyogenes were found” as “a large number of T. pyogenes was found”. Line 174
(26) We revised “The CLSM examination result illustrates that” as “The CLSM examination result illustrated that”. Line 175
(27) We revised “the biofilm of M and H luteolin groups” as “the biofilms of M and H luteolin groups”. Line 178
(28) We revised “The rats in treatment groups (M and H groups) illustrate the weight gain” as “The rats in treatment groups (M and H groups) illustrated the weight gain”. Line 185
(29) We revised “in the endometritis group (E) which indicates that the bacterial infection is serious” as “in the endometritis group (E), which indicates that the bacterial infection was serious”. Line 189
(30) We revised “was no adherence with surround-tissue and with uniform texture” as “was not in adherence with surround-tissue and uniform texture”. Line 196
(31) We revised “organs in the pelvic were adhesive and congested, the uterus volume was greatly increased and with obvious swelling and convoluted shape” as “organs in the pelvis were adhesive and congested, and the uterus volume was greatly increased with obvious swelling and convoluted shape”. Line 197-199
(32) We revised “The uterine wall is thinning, and the uterine lumen is filled with yellow pus” as “The uterine wall was thinning, and the uterine lumen was filled with yellow pus”. Line 199
(33) We revised “The swelling of the uterus as well as endometrial hyperemia and edema also can be observed” as “The swelling of the uterus, as well as endometrial hyperemia and edema, could also be observed”. Line 201
(34) We revised “only show slight hyperemia” as “only showed slight hyperemia”. Line 205
(35) We revised “(4MIC)” as “(4 × MIC)”. Line 230
(36) We revised “Endometritis is reported to partly result from Trueperella pyogenes infection” as “Endometritis was reported to partly result from T. pyogenes infection”. Line 252
(37) We revised “decreased to the standard range which demonstrates that” as “decreased to the standard range, demonstrating that”. Line 264
(38) We revised “have not returned to the reference level” as “had not returned to the reference level”. Line 266
(39) We revised “the extravasated blood was still present in the rat uterus horn. Whereas rats have im-proved conditions after the treatment of luteolin” as “the extravasated blood was still present in the rat uterus horn, whereas rats had im-proved conditions after the treatment of luteolin”. Line 268-270
(40) We revised “Luteolin can decrease the expression level” as “Luteolin could decrease the expression level”. Line 272
(41) We revised “Drug and Bacterial strain informations” as “Drug and bacterial strain informations”. Line 283
(42) We revised “purified using the Trizol agent” as “purified using the Trizol reagent”. Line 341
Reviewer 2 Report
The paper under review deals with the evaluation of the effects of luteolin on biofilm of Trueperella pyogenes and its therapeutic effect on rat endometritis. The topic of the investigation are in the scope of the journal. The evaluated paper contributes new information to the field. The study showed that luteolin could disperse large pieces of biofilm into small pices, significantly inhibit the relative expression of biofilm-associated genes luxS, plo, rbsB and lsrB and significantly reduce the symptoms of induced endometritis in rats.
This study is of interest. The results obtained may prove fruitful in future studies, contributing to the development of a new therapeutic strategy for uterine inflammation caused by T. pyogenes. The manuscript requires some corrections:
Endometritris was not induced by the T. pyogenes biofilm (lines 23, 26, 340 etc.), but by the bacterium itself. The biofilm is formed afterwards.
There are no conclusions at the end of the discussion. Please add.
Author Response
Dear reviewer,
The authors would like to thank you for your comments. Care has been taken to improve the work and address concerns as per the specific comments below.
Point 1: Endometritris was not induced by the T. pyogenes biofilm (lines 23, 26, 340 etc.), but by the bacterium itself. The biofilm is formed afterwards.
Response 1: Thank you very much for your correction and sorry for our incorrect description.
(1) We revised “a rat endometritis model was established by glacial acetic acid stimulation and T. pyogenes biofilm in this study " as " the in-vivo anti-biofilm activity of luteolin against T. pyogenes was studied using a rat endometritis model established by glacial acetic acid stimulation and T. pyogenes intrauterine infusion”. Line 23
(2) We revised “on infection caused by T. pyogenes-biofilm” as “on T. pyogenes biofilm-associated infections”. Line 27
(3) We revised “the biofilm of T. pyogenes isolate (BMH06-3)” as “a T. pyogenes isolate (BMH06-3)”. Line 253
(4) We revised “a good therapeutic effect on rat endometritis caused by the biofilm of T. pyogenes” as “a good therapeutic effect on the T. pyogenes biofilm-associated rat endometritis”. Line 261
(5) We revised “induced by T. pyogenes biofilm was lightened” as “induced by T. pyogenes was lightened”. Line 265
(6) We revised “the rat T. pyogenes biofilm-induced endometritis model” as “the rat T. pyogenes biofilm-associated endometritis model”. Line 355
Point 2. There are no conclusions at the end of the discussion. Please add.
Response 2: We are sorry for our negligence. We added " In summary, these results indicated that Luteolin is a promising anti-biofilm agent against T. pyogenes. Luteolin could significantly inhibit the expression of biofilm-related coding genes, luxS, plo, rbsB and lsrB. Luteolin also showed potent therapeutic effect on the T. pyogenes biofilm-associated rat endometritis model. This study has provided a new sight into the possible clinical application of natural-derived flavonoid compounds as well as their roles as potential candidates for anti-biofilm drug development". Line 276-281.